# $\alpha$-ReQ : Assessing representation quality by measuring eigenspectrum decay

**Kumar Krishna Agrawal**[†]
UC Berkeley
CA, USA

**Arnab Kumar Mondal**[†]
Mila & McGill University
Montréal, QC, Canada

**Arna Ghosh**[†]
Mila & McGill University
Montréal, QC, Canada

**Blake A. Richards**
Mila, Montreal Neurological Institute & McGill University
Montréal, QC, Canada
Learning in Machines and Brains Program, CIFAR
Toronto, ON, Canada

## Abstract

Self-Supervised Learning (SSL) with large-scale unlabelled datasets enables learning useful representations for multiple downstream tasks. However, efficiently assessing the quality of such representations poses nontrivial challenges. Existing approaches train linear probes (with frozen features) to evaluate performance on a given task. This is expensive both *computationally*, since it requires retraining a new prediction head for each downstream task, and *statistically*, which requires task-specific labels for multiple tasks. This poses a natural question, *how do we efficiently determine the "goodness" of representations learned with SSL across a wide range of potential downstream tasks?* In particular, a task-agnostic statistical measure of representation quality that predicts generalization without explicit downstream task evaluation would be highly desirable.

In this work, we analyze characteristics of learned representations $\mathbf{f}_\theta$ in well-trained neural networks with canonical architectures & across SSL objectives. We observe that the eigenspectrum of the empirical feature covariance $\mathrm{Cov}(\mathbf{f}_\theta)$ can be well approximated with the family of a power-law distribution. We analytically and empirically (using multiple datasets, e.g. CIFAR, STL10, MIT67, ImageNet) demonstrate that the decay coefficient $\alpha$ serves as a measure of representation quality for tasks that are solvable with a linear readout, that is, there exist well-defined intervals for $\alpha$ where models exhibit excellent downstream generalization. Furthermore, our experiments suggest that key design parameters in SSL algorithms, such as BarlowTwins [1], implicitly modulate the decay coefficient of the eigenspectrum ($\alpha$). As $\alpha$ depends only on the features themselves, this measure for model selection with hyperparameter tuning for BarlowTwins enables the search with less compute.

## 1 Introduction

The recent success of self-supervised learning (SSL) has changed the landscape of deep learning significantly. With well-engineered architectures and training objectives, SSL models learn useful representations from large datasets without relying on any labels [1, 2, 3]. Despite this progress, quantifying the *representation quality* for models trained with SSL is still an open problem. The

---

[†] These authors contributed equally to this work

36th Conference on Neural Information Processing Systems (NeurIPS 2022).

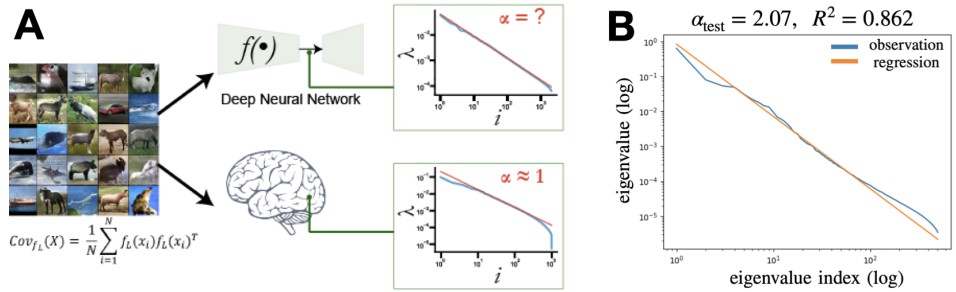

Figure 1: **(A)** One approach to evaluate representation quality is tracking the eigenspace of feature co-variance matrix $\Sigma_n(\mathbf{f}) = 1/n \sum_{i=1}^{n} \mathbf{f}(x_i)\mathbf{f}(x_i)^\top$. Analysing populations of neural activation [6, 7], suggests that the eigenspectrum for $\Sigma_n(\mathbf{f})$ can be approximated by power-law, where $\lambda_i \propto i^{-\alpha}$. **(B)** For a ResNet-50 model pretrained on ImageNet, we extract activations from an intermediate layer. We plot the eigenvalue spectrum $\{\lambda_1, ..., \lambda_d\}$ for covariance matrix $\Sigma_n(\texttt{resnet50(feats=`block4')})$ in `log-log` scale. Further, fitting a linear regressor we estimate the decay coefficient $\hat{\alpha}$.

most obvious (and common) solution is to assess performance of models using these representations on downstream tasks. However, if the goal is general representations that can be used across many domains, then this either requires a large investment of time and energy to be done well (in order to assess performance on many tasks and datasets). Alternatively, if models are assessed on only a small number of datasets and tasks, then it is hard to be confident in the assessment. Thus, we are left with a question: Can we assess the quality of learned representations without explicitly evaluating the performance on downstream tasks? To answer this question, we must formally define the "quality" of representations and subsequently examine statistical estimators that measure this property without needing downstream evaluation. Beyond theoretical interest, such a metric would be highly desirable for model selection and also useful for designing new SSL algorithms.

In search of such a metric, we turn our attention to one of the more efficient learning machines in existence – the mammalian brain. The hierarchical and distributed organization of neural circuits, especially in the cortex, provides neural representations that support a wide array of behaviours across many domains. For example, representations in primary visual cortex (V1) of mammalian brains are used by animals to support downstream behaviours ranging from object categorization to movement detection and motor control [4, 5]. This berth of downstream uses of V1 representations is desirable for artificial vision systems trained with SSL. Thus, understanding the properties of representations in V1 is a reasonable starting point for seeking a general metric of representation quality.

Recent breakthroughs in systems neuroscience enable large-scale recordings of neural activity. By recording and analyzing the response to visual stimuli, [6, 7] find that activations in the mouse and macaque monkey V1 exhibit a characteristic information geometric structure. In particular, these representations are high-dimensional, yet the amount of information encoded along the different principal directions varies significantly. Notably, this variance (computed by measuring the eigenspectrum of the empirical covariance matrix) is well-approximated by a power-law distribution with decay coefficient $\approx 1$, i.e., the $n^{th}$ eigenvalue of the covariance matrix scales as $1/n$.

Motivated by these results, we explore the use of the decay rate of the empirical eigenspectrum to characterize representation *quality* in neural networks trained with SSL (Fig. 1a). Across diverse model architectures, pretraining objectives, and downstream classification tasks, we empirically observe an eigenspectrum decay in the representation covariance matrix that roughly follows a power-law distribution (see e.g. Fig. 1b). We also find that the coefficient of this decay, denoted by $\alpha$, is informative of downstream generalization performance. Importantly, $\alpha$ can efficiently be calculated without any labels, and thereby, could be incorporated into existing SSL pipelines for efficient model selection on fixed compute budget. Our core contributions in this paper are:

1. $\alpha$ **as a potential metric** We observe that canonical pretrained architectures have representations which loosely conform to a power-law distribution in their covariance eigenspectrum. Under an assumption of a power-law distribution, we prove that the convergence rate and upper bound on generalization error are related to decay-coefficient ($\alpha$) of the corresponding power-law distribution, with best values of $\alpha$ being neither too large nor too small.

2. $\alpha$ **as a label-free measure for representation quality**: We empirically validate our theoretical results by demonstrating a relationship between $\alpha$ and downstream generalization. In particular, we find that either too high or too low an $\alpha$ value implies poor generalization, both in-distribution and out-of-distribution. Generally, the best representations are those where $\alpha$ is in a range that is close to 1, as observed in V1 of the real brain. Furthermore, these results hold irrespective of the choice of network architecture or pretraining objective.

3. $\alpha$ **for model selection in SSL**: We establish $\alpha$ as a reliable metric for model selection in a specific case of SSL, Barlow Twins [1]. Notably, we show that $\alpha$ allows us to identify model hyper-parameters that lead to representations that generalize well without any labels, more so than the actual loss function used to train the network.

Altogether, our results show that $\alpha$ is a promising task/architecture/data agnostic metric for assessing representation quality in SSL. We publicly release our results and code [†].

## 2 Theoretical Framework to Assess Representations in SSL

We are interested in evaluating the quality of high dimensional representations learned by neural networks. Formally, we consider the overparameterized setting, with datasets $\mathcal{X} \subset \mathbb{R}^d$, and learned mappings $\mathbf{f} : \mathbb{R}^d \to \mathbb{R}^D$ such that $\mathbf{f}(\mathbf{x})$ is a vector of $D$-dimensional features.

We consider DNNs as our function approximators, where each architecture implicitly defines a function class $\mathcal{F} = \{\mathbf{f}_\theta : \theta \in \Theta\}$ where $\Theta \subset \mathbb{R}^p$ is the feasible set of model parameters (e.g bounded $\Theta, [-B, B]^p$ for some $B \in \mathbb{R}$). The search for good representations poses the following optimization problem: with a dataset $\mathcal{D}_{\text{pretrain}}$ from some data distribution $\mathbb{P}_{\text{pretrain}}$ (potentially with labels), search for *optimal* parameters $\theta^*$ such that with pretrain objective $\mathcal{L}_{\text{pretrain}}(\mathbf{f}, \mathcal{D})$

$$\theta^* = \arg\min_{\mathbf{f}_\theta \in \mathcal{F}} \mathcal{L}_{pretrain}(\mathbf{f}_\theta, D_{\text{pretrain}}) \tag{1}$$

The above optimization problem is usually non-convex, and often uses gradient based optimizer to find an approximate solution $\hat{\theta}$. Typically, to measure the quality of $f_{\hat{\theta}}$, researchers evaluate the quality of representations on a downstream task $\mathcal{D}_{\text{downstream}}$ with a linear readout using some metric $\mathcal{R}(\texttt{linear}(\mathbf{f}_{\hat{\theta}}), D_{\text{downstream}})$. A concrete example is pretraining a VGG-16 model ($\mathcal{F}$) on ImageNet dataset ($D_{\text{pretrain}}$) and evaluating the learned representations using classification accuracy ($\mathcal{R}$) by learning a linear classifier on the MIT67 dataset ($D_{\text{downstream}}$). For the rest of the paper, we denote feature maps as $\mathbf{f}_\theta(x) \in \mathbb{R}^D$ where $\mathbf{f}_\theta : \mathcal{X} \to \mathbb{R}^D$, and the readout network as $\mathbf{g}_\phi : \mathbb{R}^D \to \mathbb{R}^k$, where $k$ is the target dimensionality. For simplicity of analysis, we consider linear readouts unless explicitly mentioned, i.e. $\mathbf{g}_\phi(x) = x^T \phi$.

### 2.1 Covariance estimation and eigenspectrum

For a parameterized function $\mathbf{f}_\theta : \mathcal{X} \to \mathbb{R}^D$ (assume centered), the ($n$-sample) empirical covariance matrix is defined as:

$$\Sigma_n(\mathbf{f}_\theta) = \frac{1}{n} \sum_{i=1}^n \mathbf{f}_\theta(x_i) \mathbf{f}_\theta(x_i)^T$$

The eigenspectrum of $\Sigma_n(\mathbf{f}_\theta)$ informs us about the variance explained by each principal component of the space spanned by representations $\mathbf{f}_\theta(\mathbf{x})$. Using the spectral decomposition theorem on symmetric matrices, $\Sigma = U\Lambda U^T$, where $\Lambda$ is a diagonal matrix with nonnegative entries, and $U$ is a matrix whose columns are the eigenvectors of $\Sigma$. Without loss of generality we assume that $\lambda_1 \geq \lambda_2 ... \geq \lambda_m$, where $m = \min(n, D)$ is the rank of $\Sigma_n(\mathbf{f}_\theta)$.

### 2.2 Eigenspectrum Decay in Deep Representation Learning

Recent work in characterizing representation structure in canonical DNNs has demonstrated that the covariance eigenspectrum roughly follows a power-law [8, 9, 10]. Specifically, the eigenspectrum

---

[†] https://github.com/kumarkrishna/fastssl

of a covariance matrix follows a power-law distribution $PL(\alpha)$, or *zeta distribution* if for $\lambda_j \in [\lambda_{min}, \lambda_{max}]$, the eigenvalues $\lambda_j$ are all nonnegative, and

$$\lambda_j \propto j^{-\alpha}$$

for some $\alpha > 0$. Here, $\alpha$ is the *slope* of the power law, and is referred to as the *coefficient of decay* of the eigenspectrum. Intuitively, small $\alpha$ (typically $\alpha \leq 1$) suggests a dense encoding, while a high $\alpha$ (rapid decay) corresponds to a sparse encoding.

**Insights from High-Dimensional Linear Regression** : Recent work in theoretical machine learning has connected bounds on generalization error for a linear regression problem to the eigenspectrum of the feature covariance matrix. Specifically, in the infinite-dimensional setting with $D \to \infty$, [11] studied the linear regression setting with Gaussian features, and proved that if the eigenspectrum follows a power-law distribution (up to polylogarithm factors), the min-norm solution provides good generalization performance iff $\alpha = 1$. The asymptotic regime of infinite width is an excellent framework to study theoretical properties of DNN representations. However, despite having a large number of parameters, practical DNNs always possess finite dimensional representations, making it important to investigate the implications of such results in finite width models. In particular, for the finite dimensional setting we try to answer: *Does $\alpha$ sufficiently larger or smaller than 1, still allow efficient learning and strong generalizability?*

To answer this question, we narrow our focus to gradient-based optimization techniques usually used to train DNNs. In particular, from the optimization perspective, Advani *et al.* showed that for deep linear regression in high dimensions, the time required for training and the steady-state generalization error are both $\mathcal{O}(\frac{1}{\lambda_{min}})$ [12]. A key difference from our work is that they assume the inputs are drawn from an isotropic Gaussian distribution. Instead, we investigate the generalization of linear regression on $\mathbf{f}_\theta(x)$, which has a power law structure in its covariates. We show that the training convergence time grows exponentially with $\alpha$ using the following theorem:

**Theorem 2.1.** *Let $\hat{y} = \mathbf{f}_\theta(x)^T \psi$ be an overparameterized linear regression problem where $\psi$ is learned using gradient descent in order to optimize the training error, $\mathbb{E}_{x,y}[(y - \mathbf{f}_\theta(x)^T \psi)^2]$, where $(x, y) \sim \mathcal{D}_{train}$. If we assume a power-law distribution in the eigenspectrum of representations at $\mathbf{f}_\theta$, i.e. $\lambda_n = \frac{c}{n^\alpha} \quad \forall n \geq n^*$, where $n^* \in \{1, 2...N\}$, then the time required by gradient descent to minimize the training error, $T_{convergence} = \mathcal{O}(N^\alpha)$ where N is number of training samples.*

The outline of the proof builds on key results relating to gradient descent dynamics from [13]. We show that gradient descent updates, when $\psi$ is initialized to 0, yield a recursive relation for $\psi(k)$, i.e. $\psi$ after $k$ update steps. Plugging this relation in the gradient formulation, we show that the update step length along the $n^{th}$ principal direction of $\mathbf{f}_\theta(x)$ shrinks exponentially with a decay rate proportional to $\lambda_n$. Therefore, the time to convergence in training is controlled by the smallest eigenvalue which, by design, follows the power law. In sum, Theorem 2.1 provides an explanation against arbitrarily large values of $\alpha$.

Next we show that $\alpha$ can't be arbitrarily small as the upper bound on generalization error is higher for smaller values of $\alpha$.

**Theorem 2.2.** *Let $\hat{y} = \mathbf{f}_\theta(x)^T \psi$ be a linear regression problem as before. Let us further assume that $\mathbf{f}_\theta(x) \ \forall x \sim \mathcal{D}_{train}$ is a representative subset of the inputs from true data distribution: $(x, y) \sim \mathcal{D}$. Assuming a power-law distribution in the eigenspectrum of representations at $\mathbf{f}_\theta$, i.e. $\lambda_n = \frac{c}{n^\alpha} \quad \forall n \geq n^*$, where $n^* \in \{1, 2...N\}$, the generalization error after T weight update steps, $\mathcal{G}(T)$ is:*

$$\mathcal{G}(T) := \mathbb{E}_{x,y\sim\mathcal{D}}[(y - \mathbf{f}_\theta(x)^T \psi)^2] \leq \mathcal{O}\left(r_{\hat{d}}(\Sigma(\mathbf{f}_\theta))\right)$$

$$\text{where} \qquad r_{\hat{d}}(\Sigma(\mathbf{f}_\theta)) = \frac{\sum_{i=\hat{d}}^m \lambda_i}{\sum_{i=1}^m \lambda_i} = \frac{\sum_{i=\hat{d}}^m i^{-\alpha}}{\sum_{i=1}^m i^{-\alpha}} \qquad (2)$$

*Here, $m = min(n, D)$ is the rank of $\Sigma_n(\mathbf{f}_\theta)$.*

Notably, $r_{\hat{d}}(\Sigma(\mathbf{f}_\theta))$ can be thought of as a measure of effective rank of the covariance matrix and grows with decreasing values of $\alpha$. Therefore, the upper bound for generalization error is higher for lower $\alpha$. Taken together, Theorems 2.1 and 2.2 suggest that $\alpha$ can neither be too high nor too low and there exists a tradeoff region which results in efficient learning and strong generalizability, where these representations form a good basis for gradient-based optimization on downstream task performance.

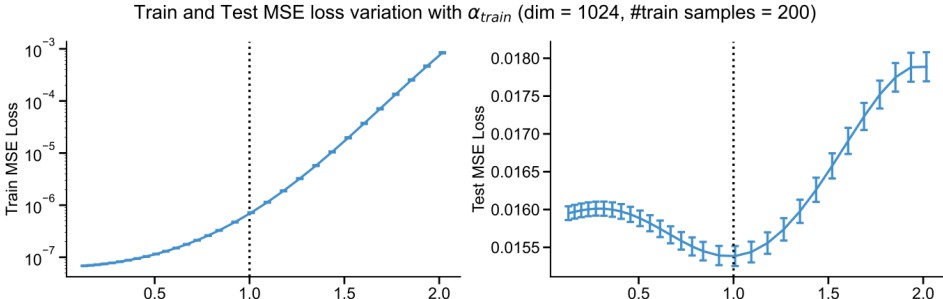

Figure 2: Overparameterized linear regression, with inputs drawn from Gaussian distribution with power-law in the covariance matrix. Models are trained with gradient descent with initialization $\psi_0 = 0$. Note that $\alpha > 1$ particularly suffers from high train, test MSE loss, with low generalization error with $\alpha \approx 1$.

**Another motivating example** Before exploring the link between $\alpha$ and generalization performance of deep networks, we validate our theorems in a simple linear regression setting.

We first consider its relationship to the finite dimensional regression setting as in [11]. Specifically, Theorem 6 of their paper states that the conditions for "benign overfitting", wherein a model can perfectly fit noisy training data without any subsequent loss of performance on testing data, may be looser in finite dimensions as opposed to the necessary and sufficient condition of $\alpha = 1$ in infinite dimensions. To empirically test this in the finite high dimensional setting, we examine linear least squares regression using different covariate structures for the input data. Formally, we consider covariates $\{x_i\}_{i=1}^N$, such that $x_i \in \mathbb{R}^d$ is sampled from a Gaussian distribution with covariance structure $\Sigma = \text{diag}\{\lambda_1, ...\lambda_d\}$ where $\lambda_j \sim PL(\alpha)$, i.e $\lambda_j \propto cj^{-\alpha}$. We assume access to the corresponding labels $\{y_i\}_{i=1}^N$ generated under a teacher function $\theta^*$, such that $y_i = x_i^T \theta^* + \epsilon_i$. We find that in this scenario there is a clear relationship between the proximity of $\alpha$ to 1 and the presence of benign overfitting. As shown in Fig. 2, when $\alpha$ is close to 1, the training loss is low, but the validation loss is also low. Thus, when $\alpha$ is close to 1, the generalization properties are at their best. This example thus provides another hint that $\alpha$ is a potential measure for how well a model will be able to generalize.

## 3 Experimental Setup

Our theoretical results suggest the following: for representations with power-law characteristics, the decay-coefficient ($\alpha$) effects the *adaptivity*, where the convergence rate of linear regression with gradient descent scales exponentially $\mathcal{O}(n^\alpha)$ with $\alpha$. On the other hand, if trained sufficiently long, the upper bound for generalization error improves with larger $\alpha$. Together, these items imply that if one wants to use gradient descent to train a downstream task then $\alpha$ for the representations used

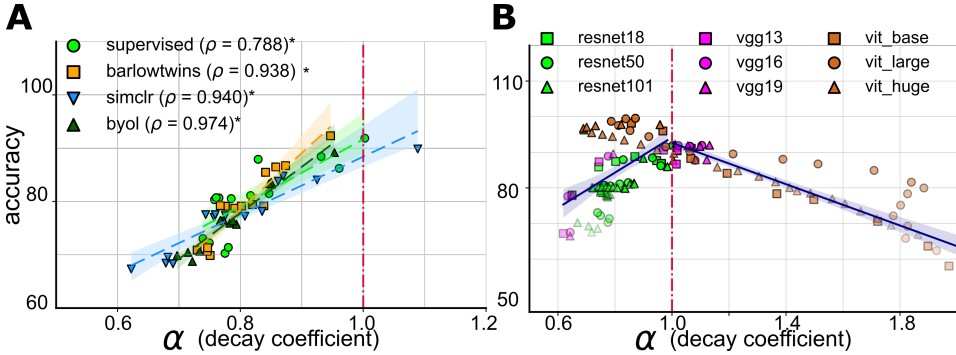

Figure 3: $\alpha$ **is predictive of out-of-distribution object recognition performance**. $\alpha$ is strongly correlated to object recognition performance on STL10 across different architectures and pretraining loss functions.

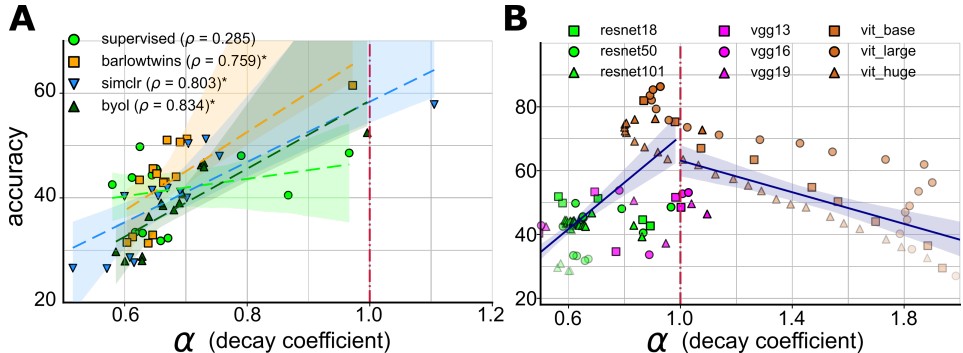

Figure 4: $\alpha$ **is predictive of out-of-distribution object recognition performance**. $\alpha$ is strongly correlated to object recognition performance on MIT67 across different architectures and pretraining loss functions.

should be neither too large nor too small, i.e. high quality representations are those with $\alpha$ in a "Goldilocks" zone. Building on these insights, we now design experiments to empirically study the potential for $\alpha$ to be used as a representation quality metric. We do so by measuring the relationship between $\alpha$ and the standard metric of representation quality, i.e. downstream task generalization performance. In this work, we restrict our scope to vision tasks, specifically object classification and scene recognition. In particular, our experiments:

- **measure the eigenspectrum decay & generalization** : We systematically evaluate *in-distribution* and *out-of-distribution* generalization, and simultaneously, we also measure $\alpha$, across diverse choices for network architectures, pretraining learning objectives, and across multiple datasets. We then look at whether $\alpha$ values near some neighbourhood around 1 are indeed predictive of representations that generalize well to downstream tasks.

- **explore $\alpha$ as a metric for model-selection in SSL:** Measuring $\alpha$ is label-free and computationally inexpensive, as it requires a single forward-pass on the downstream dataset. We thus ask, is $\alpha$ predictive of generalization capabilities for current SSL algorithms, and if so, can it be used for model selection? We run extensive ablations on design of the BarlowTwins [1] algorithm in an effort to answer this question.

### 3.1   Measuring eigenspectrum Decay & Generalization

**Evaluation Protocol:** To measure $\alpha$ for the learned representation $\mathbf{f}(x_i)$, we first extract features from intermediate layers of a DNN pretrained on $\mathcal{D}_{pretrain}$ and estimate the corresponding covariance matrix $\Sigma_n(\mathbf{f})$. Next, we compute the full set of numerical eigenvalues, and estimate $\alpha$ by a fitting a linear model on the eigenspectrum in `log-log` scale. Our pretrained models are taken from PyTorch Hub [14] and `timm` [15]. Given that we observed no significant difference between the observed $\alpha$ values in the train and test sets, we refer to this empirical estimate as the $\alpha$ for the dataset.

To estimate the capacity of intermediate representations in solving the downstream task, we train a linear readout layer, $\mathbf{g}(.)$, from representations to target logits. Intuitively, this comes down to establishing a relationship between the manifold geometry and linear separability of representations [16]. Thereafter, we observe the correlation between estimated $\alpha$ and the linear readout performance. Notably, we also tried non-linear $\mathbf{g}(.)$ and observed a similar trend in results (see Appendix B.1).

**Inductive bias of Model Architecture** In this section, we investigate the relationship between $\alpha$ of the representation covariance matrix and object/scene recognition performance in DNNs with different backbones and the role of depth. In order to do so, we examine varying depth configurations within network architectures across three generations of models on the STL10 and MIT67 dataset [17]. We choose both object and scene recognition task as they fundamentally differ in the nature of the features that must be extracted to perform well. For scene recognition, the model focuses on global features in the entire image, while for object recognition, the model focuses on local features of the image containing the object[18].

We examine deep Convolutional Neural Networks (CNNs) without any residual connection as our first family of models. Specifically, we choose three different configurations of VGG-Net [19], namely VGG-13, VGG-16 and VGG-19. We inspect representations that are input to the dropout and MaxPool layers during the forward pass of the network. Second, we consider Deep Residual Networks [20] which are widely used in computer vision. We inspect the representations that are input to each of the residual blocks as well as the Adaptive average pool in ResNet-13, ResNet-50 and ResNet-101 during their respective forward passes. Finally, owing to the recent success of transformers in object recognition tasks we consider Vision Transformers (ViT) [21] as the third family of models, namely ViT-Base/8 , ViT-Large/16 and ViT-Huge/14. Unlike VGG and ResNet, we only look at features in the intermediate layers because it summarizes the entire input image and is used in practice for class prediction. For all model architectures, we use the weights obtained from pretraining on ImageNet.

Fig. 3 and Fig. 4 illustrate the relation between performance and $\alpha$ for all the nine architectures across intermediate layer representations, as described above. We found that, while most intermediate representations in CNNs (with or without residual connections) exhibit $\alpha < 1$, representations in ViTs mostly exhibit $\alpha > 1$. Nevertheless, representations extracted from the deepest layers of all the models exhibit $\alpha$ close to 1, irrespective of the total depth of each model (see Appendix B). Furthermore, the performance on downstream task increases with depth. This is unsurprising because all networks were trained to perform object recognition on ImageNet [22] and thereby would have leveraged hierarchical processing to learn features that are tuned towards object recognition. Surprisingly enough, we observe a strong significant correlation between $\alpha$ and performance on the STL10 dataset, i.e. a different data distribution than the training dataset, across layers and model architectures ($\rho = -0.922$, $^*p < 0.05$ for representations exhibiting $\alpha > 1$ and $\rho = -0.922$, $^*p < 0.05$ for representations exhibiting $\alpha < 1$). It is worth noting here that the correlation was weaker for the earliest layers of each model. We believe that early layers learn more task invariant features that reflect the statistics of natural images [23, 24] and therefore lack task relevant information in their representations. Taken together, this observation confirms our hypothesis that $\alpha$ is a good indicator of out-of-distribution generalization performance when representations possess task relevant information. Thus, $\alpha$ has a *necessary but not sufficient* relationship with generalization (Appendix C).

**Learning objective** In this section, we first aim to understand how the $\alpha$ value changes across the layers of a fixed architecture DNN when trained with different learning objectives. We take a ResNet-50 model [20] pre-trained using three different SSL algorithms, namely SimCLR [2], BYOL [25] and Barlow Twins [1], and the supervised learning loss objectives on ImageNet-1k[22] dataset. We use a similar procedure as before to extract representations from the network and estimate $\alpha$.

Similar to results in the previous section, all networks irrespective of the pretraining loss function, exhibit $\alpha$ closer to 1 in the deeper layers in contrast to intermediate layers (see Appendix Fig. 9). This surprising result indicates that although the pre-training loss function was different, representations extracted from deepest layers are reflective of the object/scene semantics in natural images. Furthermore, Fig. 3 and Fig. 4 illustrate the strong correlation between $\alpha$ and generalization performance on STL10 and MIT67 across all pre-training loss functions. Together with results from the previous section, we validate our hypothesis that the representations that demonstrate good out-of-distribution generalization performance are characterized by $\alpha$ in the neighbourhood of 1. This suggests that high quality representations are indeed those with $\alpha$ in this region.

## 3.2  Label-agnostic metric for Model selection

With empirical evidence of $\alpha$ being a good measure for generalization, we now wish to study its suitability as a metric for identifying the best among models pretrained with SSL algorithms. A label-free metric like $\alpha$ could be beneficial when we don't have access to the downstream task annotations, and the SSL loss is not useful to distinguish models with good generalization performance (see Fig. 5). To investigate this, we exhaustively ablate the relationship between $\alpha$ and model performance across a wide range of hyper-parameters for a representative non-contrastive SSL algorithm.

Current SSL algorithms struggle with dimension collapse (characterized by large $\alpha$), due to pathologies in training dynamics. To study $\alpha$ across a wide-range of values, for our model selection experiments, we pick the *Barlow Twins*[1], a non-contrastive SSL algorithm that explicitly optimizes to avoid dimension collapse. In particular, the Barlow Twins learning objective ($\mathcal{L}_{BT}$) proposes imposing a *soft-whitening* constraint :

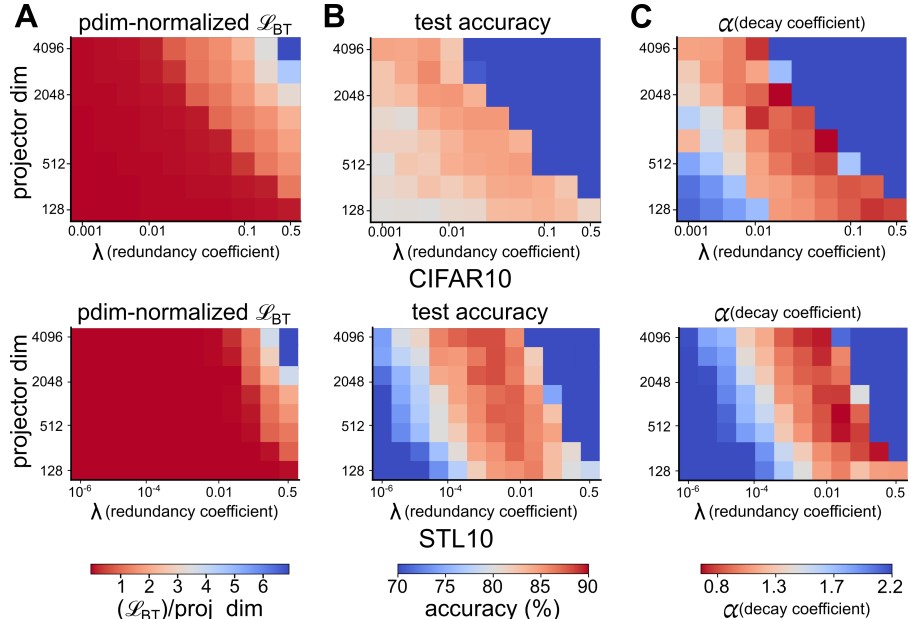

Figure 5: $\alpha$ **as a metric to inform model selection.** The SSL loss (training for same number of gradient steps) is no longer useful to distinguish models with superior downstream performance. However, decay coefficient $\alpha$ shows strong correspondence to downstream test accuracy over a large hyperparameter ranges. Measuring in-distribution generalization for (A-C) BarlowTwins trained and evaluated on CIFAR10. (D-F) BarlowTwins trained and evaluated on STL10.

$$\mathcal{L}_{\mathrm{BT}} = \underbrace{\sum_{i=1}^{d}(1 - \mathcal{C}(\mathbf{f}_\theta)_{ii})^2}_{\text{invariance}} + \lambda \underbrace{\sum_{i=1}^{d}\sum_{j\neq i}^{d} \mathcal{C}(\mathbf{f}_\theta)_{ij}^2}_{\text{redundancy-reduction}} \quad \text{s.t. } \mathcal{C}(\mathbf{f}_\theta)_{ij} = \frac{\sum_n \mathbf{f}_\theta(x^A)_i \mathbf{f}_\theta(x^B)_j}{\sqrt{\sum_n \mathbf{f}_\theta(x^A)_i^2 \sum_n \mathbf{f}_\theta(x^B)_j^2}} \quad (3)$$

Notably with sufficient large $\lambda$, the model would impose an $\alpha = 0$ constraint on the representations [26]. Despite the intuitive connection between $\alpha$ and $\lambda$, it is unclear whether this relationship holds across a wide-range of values for model hyperparameters. To empirically establish this connection, we vary $\lambda$ (redundancy coeffecient), projection head dimensionality and learning rate, and train a ResNet50 encoder using *Barlow Twins* learning objective. We provide the results for CIFAR10 [27] and STL10 [28] in Fig. 5 demonstrating that $\alpha$ is a strong indicator of in-distribution generalization performance across a large range of hyperparameter ranges of $\mathcal{L}_{BT}$. We defer the reader to Appendix Fig. 14 & Fig. 15 for similar trends in optimization hyperparameters. Furthermore, $\alpha$ is predictive of out-of-distribution generalization performance in these settings (see Appendix B.3). The same relation also holds in contrastive SSL frameworks, specifically for SimCLR [2] (see Appendix B.4).

**Model Selection on a Compute Budget:** With a constrained compute budget, using $\alpha$ for model selection is significantly cheaper than evaluating the representations on a suite of downstream tasks. To illustrate this, consider the standard alternative of training a linear classifier on the representations and evaluating test accuracy. Compared to training a linear probe, which requires multiple epochs of forward & backward passes through the training dataset's features, to achieve

Table 1: We report the compute time for CIFAR10, STL10, and ImageNet. While CIFAR10 and STL10 are trained for 200 epochs for downstream classification ImageNet is trained for 100 epochs. (tested in 1 A100)

| Dataset | w/ eigenspectrum coefficient ($\alpha$) | w/ linear probe | perf. gains |
|---|---|---|---|
| CIFAR10 (s) | $\mathbf{3}_{\pm\ \mathbf{0.2}}$ | $48_{\pm\ 7}$ | $\sim 16\times$ |
| STL10 (s) | $\mathbf{5}_{\pm\ \mathbf{0.5}}$ | $80_{\pm\ 10}$ | $\sim 16\times$ |
| ImageNet (mins) | $\mathbf{4}_{\pm\ \mathbf{0.8}}$ | $58_{\pm\ 5}$ | $\sim 15\times$ |

reasonable estimates of downstream accuracy, computing $\alpha$ requires a single PCA step on the validation dataset's features. In Table 1, we contrast the compute times for $\alpha$ and evaluating a linear probe. A detailed algorithm to use $\alpha$ for model selection is provided in Algorithm 1 and its complexity analysis in Appendix B.6.

**Algorithm 1** Model selection using $\alpha$

```
# M: Number of models that can be trained in parallel
# H: Number of sequential steps of model training
# K: Number of top models to log in memory for model selection

α_min = 0
α_max = ∞
models_dict = {}

for iter in range(H):
    # train M models in parallel, each with different hyperparam config
    trained_models = train_SSL_models(num_parallel_models=M)

    # evaluate α for the trained models
    alpha_trained_models = evaluate_alpha(trained_models)
    best_trained_models = {model: alpha for model, alpha in
        alpha_trained_models if alpha  ∈ [α_min,α_max]}
    models_stat = {model: run_linear_eval(model) for model, alpha in
        best_trained_models.items()}

    # Update α_min and α_max
    α_min = max({α_m | α_m > α_j & acc_m ≥ acc_j  ∀ m,j ∈ models_stat})
    α_max = min({α_m | α_m < α_j & acc_m ≥ acc_j  ∀ m,j ∈ models_stat})

    # Trim models_stat to keep only top K models
    threshold = get_best_models(model_statistics, topk=K)
    models_stat = {model: acc for model, acc in models_stat.items() if acc >
        threshold}

# Return best model performance from the 'selected' models
best_model_acc = max({acc_m ∀ m ∈ models_stat})
return best_model_acc
```

# 4   Related Work

**Evaluating representations and model quality** We note a substantial body of work aiming to empirically characterize the structure of emergent representations in DNN without requiring labels [29, 30]. One such index that quantifies the similarity of representations across layers (of the same or different models) is Centered Kernel Alignment (CKA) [23]. While CKA does not provide explicit guidance for downstream performance, [31] shows that the in-distribution generalization gap can be predicted using a different index based on the model's parameters. In particular, they show that the Empirical Spectral Density (ESD) of weight matrices for many DNNs obey a power-law, with the decay coefficient being predictive of in-distribution performance. In the present work, we explore similar indices that potentially correlate with out-of-distribution generalization by examining the eigenspectrum of *activations*.

**Generalization in Overparameterized Models** Modern neural networks often have significantly more parameters than the number of training samples, challenging the classical understanding of the bias-variance tradeoff. Overparameterization permits neural networks to overfit to noise in training data without impairing their generalization to unseen data. In recent work, Bubeck *et al.* proved that overparameterization is a necessary condition for smooth interpolation in neural networks [32].

Furthermore, this *benign overfitting* phenomenon in an overparameterized linear regression problem has been linked to the power law coefficient of the input covariance matrix [11]. Specifically, Bartlett *et al.* showed that benign overfitting is possible for an infinite-dimensional linear regression problem iff the eigenspectrum satisfies a power law (up to polylog factors). More recently, Lee *et al.* found that the tail eigenvalues of infinite-width network kernels exhibit a power law decay [33]. Following this, Tripuraneni *et al.* explored high-dimensional random feature regression settings and analytically showed a dependence between the eigenspectrum decay rate of the feature covariance matrix and generalization error [34]. While these characterizations provide a theoretical understanding of generalization error in the asymptotic or random feature settings, corresponding questions in the finite-dimensional DNN trained with gradient descent are open problems.

For deep linear networks trained using gradient descent, the eigenvalues of input covariance determine the generalization error dynamics [12]. Advani *et al.* demonstrated that small eigenvalues determine the convergence of training dynamics as well as the overfitting error at convergence. For overpa-

rameterized 2-layer neural networks, Arora *et al.* provided a fine-grained analysis of generalization bounds [35]. In contrast, we study modern DNN architectures and explore the covariance structure of their learned features on visual recognition tasks.

# 5    Discussion

**Summary** Our experiments suggest a strong correlation between the decay coefficient for sample eigenspectrum of representations, $\alpha$, and both in-distribution and out-of-distribution generalization performance on tasks central to computer vision.

**Representation Quality in High-Dimensional SSL** We demonstrate that assessing the quality of a pretrained model with our label-free metric ($\alpha$) is consistent with downstream generalization. While being computationally efficient, our procedure removes the dependence on labels, making the model selection more robust and amenable for privacy critical applications.

**Necessary, but not sufficient condition** Notably, a task-agnostic measure like $\alpha$ is a necessary but not sufficient condition for assessing good performance. To elaborate on this point, let us present a thought experiment. Suppose we have our ideal representation space that satisfies our claims of exhibiting alpha close to the Goldilocks zone. If we perform a set of permutation operations on individual datapoint representations, the structure of the representation space remains the same, but the mapping from representation to label is now destroyed. In doing this set of permutations, we have now arrived at a different set of representations that would exhibit the same (or similar) alpha but demonstrate a lower task performance when measured using a linear readout layer. A similar argument is presented as theorem C.1 in [36]. Extending the conclusions of this thought experiment to our observations, it is clear that we could have models with similar alpha values but distinctly different performances. But an alpha value that is not in the Goldilocks zone would be associated with inferior model performance. This property is the core of our claim and allows us to propose alpha as a measure for model selection in SSL pipelines.

**Redundancy in ViT representations** Unlike other models that we considered, representations from early layers of ViT had a rapid eigenspectrum decay with $\alpha >> 1$ (see Fig. 3 and Fig. 4). The transformer architecture has a notable difference by design, i.e. early layers possess global receptive field context via self-attention on patch embeddings. Raghu *et al.* found that early ViT layers incorporate both local and global information [30]. Based on these insights, one intuitive interpretation of our results is that the representations have a low effective rank and encode redundant information relevant across multiple scales.

**Limitations** While the role of $\alpha$ and its relationship to generalization performance is better understood in the asymptotic setting for linear regression, similar questions in finite-dimensional nonlinear models are unanswered. It is also worth noting that the empirical correlation was weaker for the earliest layers in each model. We believe that early layers learn more task invariant features such as corners and edges [23, 24], and thereby lack rich semantic information for the fine-grained downstream task. In this case, distinguishing between poorly-trained models may be inconsistent with $\alpha$.

**Future Directions** Learning efficiently at scale from unlabelled datasets poses an exciting open problem in deep representation learning. We hope this work encourages new perspectives into model selection for SSL pipelines and informs the design of learning objectives and model architectures to learn *task-agnostic, adaptive* features. Understanding the behaviour of $\alpha$ (notably, a scale-invariant metric) in high-dimensional representation learning might provide insight into developing a theoretically grounded understanding of generalization in deep neural networks.

# Acknowledgements

The authors would like to thank Zahraa Chorghay and Colleen Gillon for their aesthetic contribution to the manuscript and figures. This research was enabled in part by support provided by Mila and Compute Canada. This work was supported by Vanier Canada Graduate scholarship (AG); Healthy Brains, Healthy Lives (AG & BAR); NSERC, Grant No. RGPIN-2020-05105 and RGPAS-2020-00031 (BAR); and CIFAR, Canada AI Chair & Learning in Machines and Brains Fellowship (BAR).

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
