# OpenReview forum: "$\alpha$-ReQ : Assessing Representation Quality in Self-Supervised Learning by measuring eigenspectrum decay"
_NeurIPS.cc/2022/Conference — NeurIPS 2022 Accept_

### Official Review · Reviewer_78Rr · 2022-07-09

**Rating:** 5
**Confidence:** 4
**Soundness:** 3 good
**Presentation:** 3 good
**Contribution:** 2 fair

**Summary:**

The paper aims to find a task-agnostic statistical measure for SSL models that can serve as a good indicator of downstream performance.
With the observation that the eigenvalues of the covariance matrix of the features follow a power law, this work proposes to use the decay rate of the eigenvalue spectrum $\alpha$ (i.e. $\lambda_i \propto i^{-\alpha}$).

Theoretical analyses (Sec 2) suggest that there should be a sweetspot for $\alpha$, from trading off the convergence time and the generalization error.

Empirical study (Sec 3) are performed on STL-10 and MIT-67 datasets, with VGG/ResNet/ViT model pretrained using Barlow Twins/SimCLR/BYOL objective. The results show that there's a clear correlation between $\alpha$ and the task performance (in-distribution, or transfer).

=== **Update post rebuttal** ===

My TL;DR from the paper is that good performing models necessarily have a decay coefficient $\alpha$ that lies within a certain range, which means $\alpha$ can serve as a metric for model selection. Previously I'm concerned about the practical applicability of this metric, since empirical findings show that the dependency on $\alpha$ is unclear in practice. During rebuttal, the authors have addressed most of my concerns, in particular:
- How to find the "inflection point" of $\alpha$ for different architectures & tasks: see discussion at the end of Sec 3.2 (and Appendix B.4); the idea is that $\alpha$ can be used together with linear probe to reduce the computation budge (rather than replacing linear probe altogether, which would be impossible given that  is not a sufficient condition).
- the strength of correlation: the author addressed this here: https://openreview.net/forum?id=ii9X4vtZGTZ&noteId=XTLkW4joEFR

Overall, even though I'm still unsure how useful $\alpha$ can help on model selection (which is the main goal of the study on assessing representation quality), I think the paper provides a relatively thorough study on one plausible solution for assessing representation quality.

**Questions:**

Please note that I'm using the line numbers in the full version, i.e. paper with the appendix.

- Thm 2.2: $m$ is the rank, which may or may not equal to (i.e. could be smaller than) $\min(n, D)$?
- What's the takeaway of the training loss plot in Fig 2?
- Line 197: "no significant difference between the train & test set": for OOD, would the gap between $\alpha$ values estimated on the source and target domain be indicative?
- Line 225, about the values of $\alpha$ for CNNs and ViTs: are the dimensions kept the same across these architectures?
- Line 234 mentions that the correlation between $\alpha$ and the downstream performance is weaker at earlier layers. I wonder what this implies for the applicability of $\alpha$ as metric: can $\alpha$ be used for any representations, or only the representations from deeper layers? Are their other conditions that should be satisfied?
- If we only consider $\alpha$ values around 1, do we still see the strong correlation between $\alpha$ and the performance?
  - For example on Line 245, "$\alpha$ closer to 1 in the deeper layers": if we consider only $\alpha$ values calculated from the deeper layers, does the strong correlation still hold? In other words, are far-from-1 $\alpha$ values only from earlier layers of the network, which are rarely taken as features in practice?
- If we know that the desired value for $\alpha$ is 1, can we regularize the model in the pretraining phase to encourage the model to reach this value (at least for in-distribution generalization)?
- Line 265: please note that the Barlow Twins objective is applied on the features after the projection head, whereas for downstream tasks we take the features before the projection head, so there's a discrepancy.
- For transfer performance, are $\alpha$ values estimated on different downstream datasets similar? If different models are preferred by different datasets, is this reflected in the corresponding $\alpha$ values?

Minor points:
- Line 202, 227, 274: it would be better to specify the appendix section.
- Typos
    - Line 172: a missing closing bracket.
    - Line 245: "Appendix Fig. 3" links to the Fig 3 in the main paper.
    - Line 248: "Fig 3 and Fig 3".
    - Line 270: "Figure Fig 5".


**Limitations:**

The paper discusses the limitations and future directions. One limitation is potentially having inconsistent results between $\alpha$ and performance when $\alpha$ is based on features from early layers of a network.

There is no direct societal impact.

**Strengths And Weaknesses:**

Strength:
- The paper studies an importance question, and provides both empirical evidence and theoretical justifications.
- The paper is clearly written and easy to follow.

Weaknesses: my main concern is that it's unclear how transferrable the conclusions from the empirical results are to various different settings. It would be interesting to see whether the same trend holds; if not, then it'd be interesting to understand the conditions for which the trend holds.
- There are no results on large-scale datasets (current experiments are on STL-10 and MIT67).
- Currently strong correlations (Fig 3, Fig 5) are derived from all models, including those with poor accuracies. It's unclear whether the strong correlation still exist if we consider only relatively good performing models.
- The results in Fig 4 do not show a strong correlation; there's a weaker correlation between the model performance and the $\alpha$ value calculated from lower-layer features.

Side note: please make the theorems rigorous (at least in the appendix where there's no space constraint). For instance, please state the convergence time in terms of the error tolerance, and please avoid using $\approx$ (e.g. on line 532)

---

> ### Author Response · Authors · 2022-08-02
> **2/2 Response to reviewer 78Rr**
>
> Following the previous thread:
>
> **3. Conceptual clarifications**
> * **BarlowTwins loss applied on the projector output:** Thank you for catching this! Indeed, we missed adding the projection operator in the text. We have corrected the explanation and hope the updated version will genuinely reflect the framework and its weak relation to alpha.
> * **A key takeaway from plots in Figure 2:**  Here, we study the role of representation eigenspectrum in overparameterized linear regression. In particular, we show the models that generalize $(test_{MSE} - train_{MSE})$ best favor $\alpha$ in a "desirable range," minimizing the generalization error.
> * **m may be less than min(n, D):**  We agree with the reviewer that the covariance matrix need not be full rank. However, our theoretical results do not depend on this assumption and hold even if representations reside on a lower dimensional manifold than their ambient dimensionality.
>
> **4. Optimizing for "goodness" of representations**
>
> Designing training algorithms and learning objectives to explicitly impose "goodness" in representation space is an exciting future direction. However, we believe discussing those ideas is beyond the scope of the current work, where we hope to present convincing theoretical and empirical results to establish $\alpha$ as a reliable metric.
>
> **5. Minor Corrections**
> * **Grammar:** We thank the reviewer for pointing out several typos in the draft and have corrected them.

---

> > ### Comment · Reviewer_78Rr · 2022-08-05
> > **Thank you for the clarifications**
> >
> > Thank you for the detailed reply and clarifications.
> > My one-sentence takeaway from the paper is that good performing models necessarily have $\alpha$ lying within a certain range, which means $\alpha$ can serve as a metric for model selection.
> > However, I'm concerned about the practical applicability of this metric, since empirical findings show that the dependency on $\alpha$ is unclear in practice.
> > - It can be architecture and task dependent; for instance, ViT and CNN may prefer $\alpha$ to be smaller or greater than 1, respectively. Could you clarify how to determine the inflection point when given a new architecture/dataset?
> >   -  Also, I appreciate the comment on larger scale datasets. While I stand with the authors on concerns about accessibility, inclusiveness, and carbon footprint (plus, I don't like to be asked for larger experiments either, as we probably all do :)), the reason I asked for larger datasets is again the concern about transferability, since prior work has shown that findings on smaller datasets may not be transferable to larger ones (e.g. Sec 3.1 in Shen et al. 20: https://arxiv.org/abs/2003.05438).
> > - It's a necessary but insufficient condition: regarding the General Comment, could you point me to the line number for "the section in the Appendix that discusses at length why a task-agnostic measure like would only be a necessary, but not sufficient, condition for good performance"?
> > - Follow up on the weak correlation: could the authors please answer that if we only consider features from the usual layers (e.g. layers around the final layer of the feature extractor), where $\alpha$ are usually close to 1, do we still see a strong correlation between $\alpha$ and the performance? The reason I'm asking is that if $\alpha$ can only help remove candidates that are not used in practice anyways (e.g. early layer features), then its practical impact is severely limited.
> >
> > Thank you and looking forward to your response.

---

> > > ### Author Response · Authors · 2022-08-07
> > > **Response to reviewer 78Rr**
> > >
> > > We would like to thank the reviewer for engaging with our responses and taking the time to critically respond to most of our responses. We also appreciate the follow-up concerns raised by the reviewer and provide further clarifications/justifications below:
> > >
> > > 1. **Identifying the correct range of alpha**: In our updated Appendix, we have described an algorithm (see Section B.4) to identify the inflection point or desirable range of alpha for a particular task. Owing to the reviewer’s comments, we believe the same algorithm can be used to distinctly identify the inflection point or range of alpha for new architecture and tasks. Moreover, we have also presented an amortized complexity analysis of this algorithm to indicate the possible gains when performing hyperparameter search in SSL pipelines.
> > >
> > > 2. **Large-scale experiments**: We thank the reviewer for their understanding and sharing this study (which we were unaware of) that might indicate the lack of transferability of procedures to ImageNet experiments. However, we hope the reviewer would agree on vast literature that indicate scaling laws in neural network training (i.e scaling up the model/dataset size correlate with performance gains). Further, we would like to highlight that the models used in Figure 3, 4 are ImageNet pretrained. While we agree that results on larger downstream datasets would be desirable (we hope to explore those questions in the near future), we believe the evidence presented is compelling enough to share with the broader community.
> > >
> > > 3. **Necessary, but not sufficient condition:** We thank the reviewer for engaging in further discussions around this point. As noted in the general comment, a task-agnostic measure like alpha would be a necessary but not a sufficient condition for assessing good performance. To elaborate on this point, let us present a thought experiment. Suppose that we have our ideal representation space that satisfies our claims of exhibiting alpha close to the Goldilocks zone. If we perform a set of permutation operations on individual datapoint representations, the structure of the representation space remains the same but the mapping from representation to label is now destroyed. In doing this set of permutations, we have now arrived at a different set of representations that would exhibit the same (or similar) alpha but demonstrate a lower task performance when measured using a linear readout layer. Extending the conclusions of this thought experiment to our observations, it is clear that we could have models with similar alpha values but distinctly different performance. But an alpha value that is not in the Goldilocks zone would be associated with inferior model performance. This property is the core of our claim and allows us to propose alpha as a measure for model selection in SSL pipelines.
> > >
> > > 4. **Weak correlations:** We thank the reviewer for bringing up this point and clarifying on their initial question. We believe that we have a two-part response to their concern.
> > > Firstly, if we consider only the final couple of layers (say, last 6 layers) in each model architecture, the ResNet models trained with different loss functions still demonstrate a significant correlation between alpha and performance, when points from all loss functions are pooled together (Fig. 3a). However, when we follow the same procedure for different network architectures, all trained with object recognition cost function, we are limited by the statistical power to assess the significance of the correlation values.
> > > Secondly, we believe that the reviewer wants to understand if $\alpha$ would be a better discriminator of good vs bad representations among the candidate representations that are more commonly used in practice. In Fig 3 and 4, we show that alpha can discriminate between good vs bad representations by evaluating the correspondence between alpha and performance in intermediate layers of ImageNet-pretrained networks. In doing so, we demonstrate that inferences from using alpha are in agreement with our intuitive understanding of goodness of representations in modern deep neural networks, i.e. the representations emergent in the last layers are better for transfer to tasks that are similar to the pretraining task.
> > > Given these results, we would like to focus the reviewer’s attention to Fig. 5 wherein all representations evaluated are from the last layer of the encoder, i.e. the most pertinent layer for downstream evaluation in SSL. We show that alpha is a strong indicator of downstream performance when performing hyperparameter search in SSL whereas the SSL loss is not (please see Sections B.3 and B.4 of our appendix in addition to Fig. 5). Taken together, we believe that alpha is not merely an indicator of bad candidate representations that are anyways not used in practice but instead holds immense potential as a measure of model selection in SSL, wherein picking the best model often is computationally and time intensive.

---

> > > > ### Comment · Reviewer_78Rr · 2022-08-09
> > > > **Thank you for the further clarifications**
> > > >
> > > > Thank you for the further clarifications, I have updated my review accordingly.
> > > >
> > > > 1. Identifying the correct range of alpha: thank you for the pointer. I appreciate the discussion at the end of Sec 3.2, i.e. $\alpha$ is used together with linear probe to reduce the computation budge (rather than replacing linear probe altogether, which would be impossible given that $\alpha$ is not a sufficient condition).
> > > >
> > > > 3. Necessary, but not sufficient condition: I agree with the conclusion (e.g. the argument in the thought experiment is the same as that for the proof of Thm C.1 in Saunshi et al. https://arxiv.org/abs/2202.14037); the question was more about clarifying whether such discussion has been added to the revised draft, since I wasn't able to find it in the appendix.
> > > >
> > > > 2 & 4: thank you, these have addressed my concerns.

---

> > > > > ### Author Response · Authors · 2022-08-09
> > > > > **Thank you for updating your review**
> > > > >
> > > > > We are extremely grateful to the reviewer for being actively involved during the discussion period, taking into consideration our responses, and updating their review score. We thank the reviewer for pointing out this interesting work pertaining to contrastive representation learning and are excited that they find our work to be of acceptance quality at NeurIPS.
> > > > >
> > > > > Given that their current score is borderline acceptance, we would be curious to know if the reviewer would like to see any other changes to the current manuscript and/or appendix that would encourage them to further increase their score.

---

> ### Author Response · Authors · 2022-08-02
> **1/2 Response to reviewer 78Rr**
>
> Thank you for carefully evaluating our submission; we are glad the reviewer found our paper "well written" and "easy to follow," supported by "both empirical evidence and theoretical justifications." As mentioned, our experiments suggest a strong correlation between certain eigenstatistics and downstream performance, thereby providing an efficient label-free metric for model selection in self-supervised learning. We appreciate that the reviewer rated our submission as "3/3 Good" on Soundness and Presentation; we hope we can address the concerns raised by the reviewer over the discussion period and clarify any misreadings.
>
> **1. On "no-results on large-scale datasets"**
>
> As observed by the reviewer, we perform experiments across "STL-10 and MIT-67 datasets, with VGG/ResNet/ViT model pretrained using Barlow Twins/SimCLR/BYOL objective." Furthermore,
> * All models in Figures 3 and 4 are pretrained on Imagenet (\cite), a large-scale dataset with ~1.2M labeled images with ~224x224 resolution. Per the reviewer's suggestion, we run additional large-scale experiments to the best of our abilities, such as measuring the in-distribution generalization on ImageNet trained models. Our results overwhelmingly support our hypothesis and are consistent with the previous in-distribution tests on CIFAR10/STL10 (see Figure 5).
> * Figure 5 illustrates one of the primary contributions of this work. We hypothesize that the phenomenon of eigenspectrum decay in pretrained SSL, as presented in Figure 3, is a statistical property of representations that correlates with how well the models generalize. Across multiple orders of magnitude in the design space of this model family, such as capacity (e.g., projector-dimension 2^5-2^12) and optimization objectives (e.g., redundancy coefficient 1e-6 to 1e-1), we submit evidence in support of this hypothesis. The datasets used in these experiments are well-established and standard benchmarks (with samples in the 10^5 range) in the machine learning, computer vision community \cite{STL10 paper}.
> * A very conservative estimate of computational cost (including pre-featurization trick as suggested by reviewer eRnP) of replicating Figure 5 for Imagenet would require :
> models x (train_epoch_time x num_train_epochs + eval_epoch_time x num_eval_epochs) \approx (8x10) x (100 x 3 + 1) = 24080hrs \approx 2.75years compute time on a solid  A100 machine $\approx$ 3/hour x 24080 hrs= $72240 (non-preemptible AWS instance cost).
>
> We strongly believe that the experiments presented in the current draft, while limited in their scalability, provide evidence to support our core hypothesis. As a community, as we explore the early chapters in the frontier of deep representation learning, there are several open, challenging problems regarding transferability, model selection, etc. With the increasing scale of models, probing the properties of pretrained models pose a fundamental problem for efficient, accessible ML. In this endeavor, access to large-scale computing should not be a bottleneck in serving "a diverse and inclusive community." (c.f. [NeurIPS Mission Statement](https://neurips.cc))
>
>
> **2. Justifying significance of correlations: necessary vs. sufficiency**
>
> * **On the weakness of correlation:** We believe alpha is a strong indicator of performance across models and learning paradigms (demonstrated in Fig 3 & 4). However, the reviewer is correct in understanding that this is only necessary for good performance. In other words, while representations that exhibit good performance exhibit an alpha within the "desirable range," the reverse is not valid. The weakness of correlation in earlier layers serves as a negative control; early layers of CNNs trained on object-detection learn non-semantic information, low-level features such as edge detectors etc. We expect such models to generalize poorly to scene-classification tasks, as shown in Figure 4.
> * **On comparing alpha across datasets:** Thank you for the great question. Evaluating this is beyond the scope of our current work since this would require us to run experiments in other domains, i.e., beyond object and scene classification. Possibly have a set of networks that learn features useful for several fields but possess a differential preference for particular domains.

---

### Official Review · Reviewer_uCfg · 2022-07-09

**Rating:** 5
**Confidence:** 3
**Soundness:** 3 good
**Presentation:** 3 good
**Contribution:** 3 good

**Summary:**

The authors propose a fairly simple task-agnostic way of assessing the quality of NN representations. The idea is to evaluate the power-law exponent (alpha) of the spectrum of the empirical representation covariances produced by the network. Connecting with some recent theoretical results (that were however derived in limited settings), the authors argue that alpha close to 1 is correlated with good convergence and generalization properties. For this reason, the authors propose the extrapolation of this exponent as a possible metric of NN performance, of representation quality, and for model selection.

AFTER REBUTTAL
The authors have been very open to discussing their results and addressed most of the concerns raised by the reviewers. While in my opinion some further work is needed in order to prove that the proposed measure can truly be impactful, the present study explores an unconventional and interesting direction for developing new metrics for evaluating representations.

**Questions:**

- Shouldn't the means be subtracted when the covariances are evaluated (eq. unnumbered after (1))?
- Theorem 2.2: given that the generalization bound is not tight, is this result truly informative of the best value for alpha?
- The results shown in figure 3 seem to indicate the claimed importance of alpha=1 only qualitatively, but the behavior is not at all clean. How can alpha be used for model selection or performance assessment if the power-law behavior is only an approximation, and if the extrapolated exponents can hover around 1 with alternating perfromances?
- Plots in figures 3-4 are very crowded and the descriptions are too far from their appearance.
- line 232-233: there seems to be a typo in the correlations, getting the same value on the two sides seems very unlikely.
- In figure 5, the "connection" between alpha and the test error is very hard to assess in this way. The plots are coarse-grained and the color map is unhelpful (wouldn't be better, for alpha, to assign a specific color to 1, so that the "ideal" region is highlighted?). Is the relationship still just qualitative, or is it exactly true that the closer one gets to 1 the better it is? If not, how can this observation become useful in practice?
- Did the authors try to explicitly enforce the power-law behavior with the desired exponent at training, even on a small model? Is it possible to actually test the thesis presented in this paper in a more scientific way?

**Limitations:**

The limitations of this work seem to reside in its very qualitative nature. In my opinion, the authors do not address nor acknowledge this issue, and the work would need a major revision in order to accommodate for it.

**Strengths And Weaknesses:**

The idea of looking at the spectra of empirical representation covariances seems in line with a series of recent theoretical works and seems to explore an interesting research direction.
However, what the authors propose in practice is based on a very strong assumption on the eigenvalue distribution, which admittedly does not hold in general. Moreover, the presented numerical results do not show a clear quantitative behavior, aligned with the authors' speculations, but just a qualitative agreement. If this method is to be seriously employed for model selection, I believe a more scientific inquiry, where the authors try to disprove their assumptions, could really help shed some light on the very complicated phenomenon of representation learning.

---

> ### Author Response · Authors · 2022-08-02
> **2/2 Response to reviewer uCfg**
>
> **4. Optimizing for $\alpha$**
> Explicitly enforcing a power law exponent  Designing training algorithms and learning objectives to impose "goodness" in representation space explicitly is an exciting future direction. However, we believe discussing those ideas is beyond the scope of the current work, where we hope to present convincing theoretical and empirical results to establish $\alpha$ as a reliable metric.
> Deriving the optimal $\alpha$ value from theorem 2.2  We want to thank the reviewer for bringing up this connection between Theorem 2.2 and optimizing for the alpha value that yields the best task performance. However, while Theorem 2.2 indicates an inflection point in the relationship between alpha and performance, finding a closed-form solution for alpha just from the upper bound might be infeasible. Furthermore, the optimal value of alpha depends on the properties of data and task, e.g., SNR of the targets, which might be unknown in practice. Therefore, we propose identifying the optimal value of alpha empirically. We explain this algorithm in the Appendix in the context of model selection in SSL pipelines. Thanks to the reviewer's comments, we will now add a pointer to this when we describe Theorem 2.2 to make its utility and limitations clear to the reader.
>
> **5. Minor Corrections**
> * **Grammar:** We thank the reviewer for pointing out several typos in the draft and have corrected them.
> * **Plot presentation:** We have reorganized the plots in Fig 3,4 by defining abbreviations and providing a scaled version in the Appendix.
> Mean removed while computing covariance: Indeed, we center the features before computing the covariance matrix. We will update the equation in the manuscript.

---

> > ### Comment · Reviewer_uCfg · 2022-08-06
> > **Response to the Rebuttal**
> >
> > I would like to thank the authors for their answers and for their efforts to clarify and improve the manuscript.
> >
> > Unfortunately, at this time I am still a bit skeptical about the usefulness of these observations, especially for assessing representation quality, when the proposed metric does not guarantee good generalization performance and when the range of optimal values is not so well defined. Since similar concerns were raised by reviewer 78Rr, I will wait and see how the authors respond to his questions before reassessing my score.
> >
> > Thank you!

---

> > > ### Author Response · Authors · 2022-08-07
> > > **Response to reviewer uCfg**
> > >
> > > Thank you for taking the time to review and engage with our work. We really appreciate the feedback, and will work on improving the presentation to incorporate these comments.
> > > 1. **Utility of $\alpha$ for assessing representation quality**: We would like to bring the results presented in Figure 5 to the reviewer’s attention. Beyond the utility for model selection, we’d like to highlight that our results suggest label-free measures to understand and compute the “goodness” of self-supervised learned representations.
> > > 2. **Guarantees of generalization**: As recommended by the reviewer, we have included a discussion (please see Appendix C) on the necessity (rather than sufficiency) of alpha as a metric to understand in-distribution and out-of-distribution generalization. Furthermore, we would like to point out that our theorems suggest an upper bound on generalization error with linear readouts.
> > > 3. **Optimal values for alpha**: The optimal values of alpha indeed depend on the dataset, architecture and providing a precise characterization of that is beyond the scope of this work. However we do include an algorithm (please see Appendix section B.4 in the updated supplementary) to find an optimal model efficiently by searching for the range of alpha that exhibits good downstream generalization.

---

> > > ### Author Response · Authors · 2022-08-09
> > > **Concerns of reviewer 78Rr addressed**
> > >
> > > Dear Reviewer uCfg,
> > >
> > > In response to all of the reviewer comments, we have made a number of changes to the manuscript to better demonstrate the usefulness of our proposed metric, $\alpha$, in assessing representation quality for model selection. We would especially like to point out our efforts in addressing the concerns you shared with reviewer 78Rr (please see https://openreview.net/forum?id=ii9X4vtZGTZ&noteId=XTLkW4joEFR), which the reviewer commented they found to be satisfactory and have accordingly raised their score by 2 points. We hope that you will similarly find our manuscript to be of satisfactory quality and reassess your score accordingly.
> > >
> > > Paper 7255 Authors

---

> > > > ### Comment · Reviewer_uCfg · 2022-08-09
> > > > **Reply**
> > > >
> > > > I thank the authors for their efforts. I believe many points have been clarified and that most of my concerns were addressed, at least partially, in this revision process.
> > > >
> > > > I don't feel confident to increase the score too much, but I think a negative score is not deserved at this point. I will thus increase my score to a 5, which should allow its acceptance in the conference.

---

> > > > > ### Author Response · Authors · 2022-08-09
> > > > > **Thank you**
> > > > >
> > > > > We would like to thank the reviewer for appreciating our efforts and identifying the merits of our paper. We are also grateful to them for increasing their score beyond the acceptance threshold. In line with the reviewer's comment, we too hope that our paper gets through and we get an opportunity to present our results and findings to broader NeurIPS community.

---

> ### Author Response · Authors · 2022-08-02
> **1/2 Response to reviewer uCfg**
>
> We thank the reviewer for their insightful feedback. While we incorporate several suggestions and improve the presentation of some experiments, we respectfully disagree with the assessment that our arguments are merely "qualitative" and lacking in "scientific inquiry." We believe this interpretation is due to possible misunderstandings. Based on the reviewer comments, we will update our presentation and hope to clarify any concerns over the rebuttal and discussion period.
>
> **1. Clarification of empirical evidence and assumptions for eigenspectrum analysis**
>
> Our core hypothesis examines the structure of the eigenspectrum in well-trained neural networks. In particular, we examine how well the family of power-law distributions approximate this structure and whether the decay coefficient of such distributions informs us about the generalization (in-distribution and out-of-distribution).
>
> To test this hypothesis, we consider state-of-the-art models pretrained on ImageNet. We observe that for well-trained neural networks on image classification, across multiple canonical architectures, datasets, or learning objectives, the emergent representations share a common characteristic in their eigenspectrum: a signature decay in the eigenvalues. Quantifying the best power-law fit, we obtain an $R^2$ coefficient $\geq 0.85$. Based on this evidence, we infer that power laws are a good model for the observed eigenspectra. Furthermore, we establish that empirical estimates of alpha are indeed a good measure of eigenspectrum decay.
>
> Our second hypothesis explores the effectiveness of $\alpha$ as a metric for model selection in transfer learning. Focusing on the paradigm of self-supervised learning (SSL), in figure 5, we empirically verify the ubiquity of this phenomenon across multiple orders of magnitude in the design space of non-contrastive SSL algorithms (BarlowTwins).
>
> We concur with the reviewer that generalizing this to the generic class of nonlinear neural networks would be an exciting research direction; in this work, we present a step in that shared interest, a formal hypothesis, and substantial evidence to support it in controlled experimental settings. Finally, we will update the manuscript to clarify the assumptions and limitations of our modeling assumptions; thank you for highlighting this!
>
> **2. Using $\alpha$ for model selection**
>
> We hypothesize that $\alpha$ is a strong indicator of performance across models and learning paradigms, and provide evidence to support this claim, as demonstrated in Fig 3 & 4. However, the reviewer is correct in understanding that this is only necessary for good performance. In other words, while representations that exhibit good performance exhibit an alpha within the "desirable range," the reverse is not valid. We elaborate on this issue in a general comment and update the manuscript to include a further discussion of this point. Furthermore, in the Appendix, we have laid out a concrete algorithm for model selection in SSL pipelines and its average time complexity. We believe that these updates to the manuscript will give the reader a better understanding of how to leverage alpha for model selection in practice.
>
> **3. On the correlation between $\alpha$ and generalization**
>
> * **Lack of quantitative behavior in correlation plots:**
> We thank the reviewer for this helpful comment. We agree that marking "1" as the choice of the inflection point (which is data-dependent) in the correlation plots could seem somewhat ad hoc to the reader. As noted above, our theoretical results do not indicate that the inflection point should be 1. Instead, it should depend on the data and task properties. We will update the plots by computing the inflection point from the observed data points (for Fig.3, 4) and subsequently calculate the correlation between task performance and alpha on either side of the inflection point. We hope that the reviewer finds this change to be a more quantitative description of the relationship that also supports our proposal to use alpha as a metric for model selection in SSL pipelines.
> * **Hard to evaluate correlation values in Fig 5:** Thank you for the suggestions on improving the visualizations in Figure 5. Our intent for Figure 5 is to map the performance and loss landscape across diverse hyperparameter configurations and understand the advantages of using alpha in model selection. We will include the correlation plots between alpha, and performance and explain how a model selection algorithm could use alpha as a metric.

---

### Official Review · Reviewer_eRnP · 2022-07-11

**Rating:** 6
**Confidence:** 3
**Soundness:** 1 poor
**Presentation:** 3 good
**Contribution:** 3 good

**Summary:**

The paper aims at finding a way to measure the quality of the representations learned in self-supervised learning (SSL), without having to evaluate them on downstream tasks. Based on recent experiments concerning the variance of neural representations in mammal brains, they consider the decay of the eigenspectrum of the representations covariance to evaluate those representations.  Through theoretical results, they suggest that the decay should be neither too quick nor too slow to enable downstream predictors that can be trained quickly but generalize well. Experimentally, they seem to suggest that the best representations always have a similar decay and this decay can thus be used as a metric for SSL. Furthermore, this decay is close to the one previously found in mammal brains.

**Questions:**

-  what are the blue lines in figure 3b and 4b? are they linear regressions on every point before and after $\alpha=1$? in particular does the left ones take into account the ViT models on the left of $\alpha=1$, which show the inverse correlation?
- how are you training the linear predictor? Are you encoding the images once before training or at every epoch?

**Minor suggestions/mistakes**(no need to answer)
- please use \citet when the reference is part of the sentence
- line 46: the citation for [7] should be [8] I believe, ie, you cite the wrong article.
- line 75: agnostic and metric -> agnostic metric
- line 84: With -> with
- equation 1: should be $f\_{\theta^*} = \dots$
- line 93: is target -> is the target
- line 96: why do you not remove the mean feature when computing the empirical covariance matrix? I.e. this is not the empirical covariance matric, right?
- line 110: "Regression : Recent" -> "Regression: recent"
- line 114 : upto -> up to
- line 227: "all the models exhibit \alpha,$ is unclear
- line 270: Figure Fig. -> Fig.
- line 273: Furthermore, there is $\alpha$ -> Furthermore, \alpha
- figure 5: why are there missing values (in white)?




**Limitations:**

The main limitations of the proposed method are that it seems to be: (1) highly dependent on the architecture; (2) slow despite that being the original motivation.

The former is somewhat discussed in the paper. The latter is not sufficiently well discussed assuming that linear probing is much quicker than suggested in Table 1 (see weaknesses above).

**Strengths And Weaknesses:**

**Update**: As pointed out by all reviewers (me included) I think that there are valid concerns about using the metric proposed by the authors for selecting SSL representations. For me, the main contribution of the paper is not that much about having some new metric for model selection, but rather providing experimental and theoretical insights into understanding what constitutes useful SSL representations.

---

**Strengths**:
- **Well written**: the paper is clear and was a pleasure to read.
- **Problem well-motivated and of interest** the goal of evaluating representations efficiently is of interest to the ML community.
- **These insights could help to understand and design SSL**: even if the method is never applied as a metric for SSL, I think that it sheds light on important properties of desirable representations that could be of interest for designing or theoretically understanding SSL algorithms.
- **Connects ML and neuroscience** this paper sheds light on the recent findings in neuroscience that could be of interest to the representation learning community. These findings are the starting point of the paper (not a contribution) but disseminating these interesting findings to the ML community has its own value.

**weaknesses**
- **Doubts about computational gains** the main selling point of the method is its computational efficiency, yet it is not clear that there are major advantages compared to linear probing. In particular, the large computing time for linear probing (26 hours for ImageNet) makes me doubt those results. My guess is that you are not prefeaturizing the representations, ie, encoding each image only once before training and then saving the features. In my experience prefeaturization + training a linear predictor takes much less than an hour on a single GPU and 16-bit precision. Please correct me if I'm wrong, but if this is the case, then the advantage of using $\alpha$ as a metric is much less clear.
- **Only useful for a given architecture** In figures 3 and 4 (a and b) performance of representations extracted by CNNs seem to always be strongly positively correlated with $\alpha$ while $ViT$ are inversely correlated. If one was to use $alpha\approx 1$ for choosing representations in ViTs you would be quite wrong (around 10 accuracy points off). This suggests that one can only use $alpha$ to compare representations learned using similar encoders.
- **Some experimental evidence is sometimes misleading**  I find figures 3 and 4 to be somewhat misleading, for example for each architecture there is no sign of an inflection point around and so one can change the value of $\alpha$ by changing the ratio of CNNs vs ViTs. Furthermore, the plots and correlations are somewhat arbitrary, eg, in figure 4b if you plotted and computed the same correlations for $alpha =0.8$ or $0.9$ you would get high correlations (maybe even higher than currently). Table 1 is also misleading if no prefeaturization is applied.

---

> ### Author Response · Authors · 2022-08-02
> **Response to Reviewer eRnP**
>
> We thank the reviewer for their thoughtful feedback, concisely outlining our paper's contributions. The insights and critique on experiment design for measuring computational gains helped us immensely, translating to performance improvements of our baselines and proposed estimators for model selection in SSL. Below, we clarify the scope of our contributions and address the critical concerns on technical soundness with additional experiments and discussion.
>
> **1. Correction to claims on the computational efficiency of alpha:**
> We thank the reviewer for the handy suggestion of pre-featurization, aka caching the features before training the readouts to evaluate task performance. We implemented this protocol, drastically reducing the time to prepare linear readouts on pretrained representations. We present our updated results in table 1.
> To elaborate on the updates, we first explain the three types of datasets used in each of our experiments:
>  (a) pretrain-set $D_{pretrain}$   (b) finetuning train-set $D_{train}$  (c) finetune evaluation-set $D_{evaluation}$
> Note that $D_{pretrain}$ is used for training the backbone. Thereafter, we compare the computational efficiency of our proposed metric $\alpha$ with linear-probe evaluation on a downstream task. The task-specific train/validation sets are assumed to be drawn from same distribution.
>
> Specifically, the linear probe is trained on with  $D_{train}$ (using cached features). Subsequently, we compute the accuracy of the probe on ${D}_{evaluation}$. Notably, $\alpha$ is estimated efficiently using label-free validation dataset, instead of the entire dataset [e.g. (10000 images vs 60000 (image, label) pairs for CIFAR10].
>
> While prefeaturization improves the linear-probe time-complexity, we still achieve about **10x speedups** (Table 1) by using $\alpha$. Again, we thank the reviewer for their astute observations and believe that the updated numbers better reflect the expected computational gain.
>
> **2. Role of architecture and model family on correlation:**
> A core contribution of our work is revealing the difference in eigenspectrum characteristics of well-pretrained convolution-based CNNs (ResNets) compared to transformer-based models (ViT). We believe our insights complement recent empirical observations in the machine learning literature (c.f. [Raghu etal 2021](https://arxiv.org/abs/2108.08810)).
>
> Intuitively, the lens of inductive bias in model architectures hints at a plausible explanation of this phenomenon. Notably, well-tuned patch-based ViTs capture global information across all layers of information processing and are known to encode dependencies at different scales at each layer. Therefore, each layer must encode redundancy in its representation. In contrast, pretrained CNNs have a receptive field that monotonically increases with depth, learning local features in early layers (e.g. edges). Hence, the best CNN models (we chose models pretrained on Imagenet, i.e. near state-of-the-art) learn representations that exhibit $\alpha$ < 1, whereas ViTs exhibit representations that mostly show $\alpha$ > 1. We believe these factors explain the lack of an inflection point within intermediate representations of a single model.
>
> Next, we fix the model architecture and test whether $\alpha$ serves as a useful metric for hyperparameter selection in ssl. Using a ResNet50 backbone (Fig 5), we find that the same model architecture (CNN) could learn representations that exhibit a wide range of alpha values (0.5, 1.8). Across several ablations, we observe a well defined “desired zone” of alpha, where models exhibit low generalization error.
>
> Finally, we would like to emphasize that our results suggest that alpha is a necessary condition for good performance but not sufficient. In other words, good models have $\alpha$ in the inflection range, but not all models in this range generalize well. We have added a general comment on this issue and included a discussion in Section 5 of the manuscript.
>
> **3. Experimental details and Eigenspectrum Computation:**
> * **Choice of 1 as the inflection point (figure 3, 4):**
> We agree with the reviewer; one could easily misinterpret the dashed line at 1 in Fig 3, 4 as the empirical value of $\alpha_{inflection}$, where representations yield the best downstream performance. We will update our plots by estimating $\alpha_{inflection}$ to reflect the dependence on underlying task distribution instead of universally setting this to 1.
> * **Interpreting results and plots (figure 3, 4):** Indeed, in Figures 3, and 4, the blue lines indicate the correlation separately for models in the low-alpha (below $\alpha_{inflection}$) and high-alpha regimes.
>
> **4. Minor Corrections:**
> * **Grammar/Typo:** Thank you for catching these errors; we've fixed these in the latest draft of the paper.
> * **Mean centering for the covariance matrix:** The features are mean-centered for computing the covariance matrix; we've corrected the definition.

---

> > ### Comment · Reviewer_eRnP · 2022-08-06
> > **Thank you for your answers, I'm keeping my score.**
> >
> > Thank you for answering my questions. I have read the rebuttal and other reviews. I still think that the paper provides interesting insights and could be of interest to the community but some claims could be better supported experimentally. Furthermore, as all the reviews pointed out, there are valid concerns about using $\alpha$ to measure the quality of a representation. For me, the main contribution of the paper is not that much about having some new metric for model selection, but rather providing experimental and theoretical insights into understanding what constitutes useful SSL representations. I thus wonder whether making such a contribution more prominent would be useful for future readers.
> >
> > 1. **Computational efficiency**
> >
> > Thank you for running the more fair baseline and making the changes in the paper. The computational time for linear probing now seems to be the right order of magnitude.
> >
> > That being said, I still think a computationally bounded practitioner would prefer training a linear predictor. E.g., a good linear predictor can easily be trained with $\leq50$ K examples instead of 1M, which would then be even quicker than your method and a much better estimate of the downstream performance. So I don't think that computational efficiency on a single task is the main advantage of your method but rather the unsupervised nature of it and other interesting insights from your paper. I would suggest the authors have such a discussion in the paper so that future readers do not dismiss it as disingenuous. Not having such discussion is probably fine as long as you do not make extremely strong claims on your method being more efficient on a single task than linear probing (right now is arguably borderline).
> >
> > 2. **The impact of architecture**
> >
> > I have read the rebuttal, but I don't think that it convincingly answered my concern. For example FIgure 3b (similar for Fig 4) shows that the best ViT have $\alpha \approx 0.7$ and show no inflection point even though there are many ViT with $\alpha \leq 1$. I'm definitely less concerned with the ResNet architecture.
> >
> > Given the above results, I don't quite understand what the authors mean by the following
> > > In other words, good models have $\alpha$ in the inflection range, but not all models in this range generalize well.
> >
> > In particular, what does that mean for an architecture that has no inflection point?
> >
> > I also don't quite agree with the following as there are many ViTs with $\alpha \in [0.7,1]$ in Fig 3.
> > > whereas ViTs exhibit representations that mostly show  > 1. We believe these factors explain the lack of an inflection point within intermediate representations of a single model.

---

> > > ### Author Response · Authors · 2022-08-09
> > > **Clarifications on interpretation of results, contribution claims**
> > >
> > > We thank the reviewer for their insightful comments. We agree that highlighting and elaborating the discussion on "good representations" and the observation of an "inflection point" in the model landscape would be helpful for future readers. We will revise the draft to incorporate these suggestions. Meanwhile we address some of the remaining concerns below:
> > >
> > > 1.**Computational efficiency:** We are glad to have resolved most concerns on the empirical contributions, establishing $\alpha$ as a far more efficient metric for measuring representation quality, particularly in self-supervised learning. Training linear probes with less samples would definitely be another strong baseline. However, we hope the reviewer would agree that settings where access to annotated samples is expensive, such label-free metrics could be particularly useful. As recommended, we will improve the discussion to clarify that $\alpha$ is indeed a *complementary (label-free) metric*, not a substitute, for linear probe evaluation and informs us about *useful SSL representations*.
> > >
> > >
> > > 2.**Impact of architecture:** Thank you for flagging these concerns and excellent followup questions. As the reviewer promptly observes, establishing a reasonable correlation requires presenting evidence across a considerable interval of the intended metric $\alpha$ and different canonical real-world settings. In this context, please recall that the different markers on each plot for figure 3, and 4 correspond to linear-readout performance for features from different layers of a well-pretrained vision model. We apologize for some miscommunications regarding the interpretation of these results, we elaborate on these points further:
> > > > *"whereas ViTs exhibit representations that mostly show > 1."*
> > >
> > > By suggesting that ViTs have most layers in $\alpha$>1, we intended to communicate the stark contrast in the range (0.65, 2) of $\alpha$ for Transformer architectures (ViT), where the well-trained CNNs (ResNet) concentrate in the range [0.6, 1.0]. We believe this observation is consistent with prior observations on similarities, differences in representation learning for ResNets/ViTs [Dosovitskiy etal, 2020](https://arxiv.org/abs/2010.11929), [Raghu etal, 2021](https://arxiv.org/abs/2108.08810).
> > >
> > > > what does that mean for an architecture that has no inflection point?
> > >
> > > Please note that in Figure 3/4, we look at internal representations of a fixed, well-trained vision model. As with ResNets, the $\alpha$ for such features doesn't necessarily cover the "inflection range.", i.e. we don't observe the range for *peak* $\alpha$. In figure 5, we provide extensive empirical evidence for the existence of the "inflection range" in ResNets (on CIFAR/STL) by training and evaluating models across a wide range of $\alpha$.
> > >
> > > > "We believe these factors explain the lack of an inflection point within intermediate representations of a single model."
> > >
> > > Apologies for the loose wording. The statement was intended only for ResNet models, where we observe a monotonic improvement of performance with alpha consistently. On careful examination of figures 3 and 4 for the ViT architectures (e.g., vit-huge), the reviewer would observe that there is indeed an inflection point, where model accuracy peaks at around $\alpha=0.85$. We include a higher-resolution version of all plots in Figures 3, and 4 in the Appendix (Appendix D).
> > >
> > > > In other words, good models have $\alpha$ in the inflection range, but not all models in this range generalize well.
> > >
> > > In figure 3B, the reviewer may note that there are models for ViT which have similar values of $\alpha$ but have a pretty significant gap in downstream performance. As such, we hypothesize that all models with good downstream generalization fall in the *inflection range* (IR). Further, there exist models that exhibit $\alpha \in \mathrm{IR}$, but generalize poorly. Therefore, $\alpha \in \mathrm{IR}$ is a necessary condition for "goodness of representations", not sufficient.
> > >
> > > Finally, we thank the reviewer for actively engaging and helping us improve the presentation of this work. We agree that further large scale experiments could help solidify the scope of our results. Are there any particular experiments that the reviewer thinks will help strengthen the claims and contributions? Thank you for taking the time; we really appreciate it.

---

### Author Response · Authors · 2022-08-02
**General comment**

We thank all the reviewers for their valuable comments that have helped us improve certain aspects of the manuscript and we hope that the reviewers will find the updated version to be a more convincing presentation of our claims. We realized that a common concern among all reviewers was the understanding of the extent of the relationship between $\alpha$ and downstream task performance. Through this comment, we would like to indicate the following changes to the manuscript that will hopefully clarify the nature of the observed coupling between $\alpha$ and performance:
1. Having a value of $\alpha$ close to the sweet spot or "desirable zone" is a **necessary condition for good performance, but not sufficient.**
2. We have now edited the main text to highlight this point by noting the caveats of the assumptions imposed in deriving our theoretical results. Moreover, we have added a section in the Appendix that discusses at length why a task-agnostic measure like $\alpha$ would only be a necessary, but not sufficient, condition for good performance.
3. While discussing the results, we have explicitly highlighted those results that indicate this necessary, but not sufficient, nature of the relationship.

We believe that in doing so, the reader will understand the challenges of defining a task-agnostic measure like alpha for model selection in SSL pipelines and be able to appreciate the merits of our contribution and model selection algorithm.
Firstly, we observe that alpha close to the Goldilocks zone is a necessary condition for good performance, i.e. representations that demonstrate good task performance are characterized by an alpha value close to the Goldilocks zone. However, the reverse is not true, i.e. representations characterized by a desirable alpha value do not always exhibit good task performance. Note that alpha depends only on the structure of the representation space and does not depend on the task labels. Therefore, certain representation space could have the desired structure but the semantic information about the labels might not be preserved in this space, thus owing to poor task performance measured using a linear readout. In our experiments, we note that inferences about linear readout performance also extend to performance measured using shallow non-linear multi-layer perceptrons. Therefore, we will proceed to elaborate on this subtlety of the alpha-performance relationship using the setup of our theoretical results.

In our theorems, we assume that there exists some ground truth linear weights, also termed as teacher weights, that map the representations to their targets. We measure convergence and generalization error with respect to this set of teacher weights. Therefore, our theorems do not indicate performance on the absolute scale but instead relative to the performance of the teacher weights.

---

### Meta-Review · Area_Chair_uJxw · 2022-08-23

**Recommendation:** Accept
**Confidence:** Less certain

**Metareview:**

Decision: Accept

This paper proposes a task agnostic method to evaluate the representation quality of neural network, by looking at the eigenspectrum decay power factor for the feature covariance. Theoretical results and empirical results show the method can be used as a cheap alternative for understanding neural networks and performing model selection.

Reviewers commended the approach being simple yet novel, and they are happy with the overall clarity of the manuscript. The main criticisms were:
1. Whether experimental results generalize to larger dataset and network architectures.
2. The interpretation of the method: useful tool for model selection or for understanding neural networks only?

In author feedback the above questions were partially addressed. Note here the lack of big data experiments is not a major factor of my decision. The more important point is the authors’ clarification on the eigenspectrum decay being “necessary condition” for good performance, which makes the argument for model selection weaker in a theoretical sense, although the empirical results seems to be ok. I would suggest the authors to make a clear discussion on this as promised in author feedback.

As a side note, my brief read of the paper seems to tell me that the theory part considers pre-training, and I would say pre-training is a broader concept than SSL. This means the theory is not specific to SSL, and I hope the authors can clarify this point.

Also I suggest the authors to discuss He and Ozay ICML 2022 in light of this paper’s results.
https://proceedings.mlr.press/v162/he22c.html

**Award:**

No

---

### Decision · Program_Chairs · 2022-09-14

Accept